# Overcoming Multi-model Forgetting

## Abstract

We identify a phenomenon, which we refer to as *multi-model forgetting*, that occurs when sequentially training multiple deep networks with partially-shared parameters; the performance of previously-trained models degrades as one optimizes a subsequent one, due to the overwriting of shared parameters. To overcome this, we introduce a statistically-justified weight plasticity loss that regularizes the learning of a model's shared parameters according to their importance for the previous models, and demonstrate its effectiveness when training two models sequentially and for neural architecture search. Adding weight plasticity in neural architecture search preserves the best models to the end of the search and yields improved results in both natural language processing and computer vision tasks.

## 1 Introduction

Deep neural networks have been very successful for tasks such as visual recognition (Xie & Yuille, 2017) and natural language processing (Young et al., 2017), and much recent work has addressed the training of models that can generalize across multiple tasks (Caruana, 1997). In this context, when the tasks become available sequentially, a major challenge is *catastrophic forgetting*: when a model initially trained on task A is later trained on task B, its performance on task A can decline calamitously. Several recent articles have addressed this problem (Kirkpatrick et al., 2017; Rusu et al., 2016; He & Jaeger, 2017; Li & Hoiem, 2016). In particular, Kirkpatrick et al. (2017) show how to overcome catastrophic forgetting by approximating the posterior probability, $p(\boldsymbol{\theta} \mid \mathcal{D}_1, \mathcal{D}_2)$, with $\boldsymbol{\theta}$ the network parameters and $\mathcal{D}_1, \mathcal{D}_2$ different datasets representing the tasks.

In many situations one does not train *a single model for multiple tasks* but *multiple models for a single task*. When dealing with many large models, a common strategy to keep training tractable is to share a subset of the weights across the multiple models and to train them sequentially (Pham et al., 2018; Xie & Yuille, 2017; Liu et al., 2018a). This strategy has a major drawback. Figure 1 shows that for two models, A and B, the larger the number of shared weights, the more the accuracy of A drops when training B; B overwrites some of the weights of A and this damages the performance of A. We call this *multi-model forgetting*. The benefits of weight-sharing have been emphasized in tasks like neural architecture search, where the associated speed gains have been key in making the process practical (Pham et al., 2018; Liu et al., 2018b), but its downsides remain virtually unexplored.

In this paper we introduce an approach to overcoming multi-model forgetting. Given a dataset $\mathcal{D}$, we first consider two models $f_1(\mathcal{D}; \boldsymbol{\theta}_1, \boldsymbol{\theta}_s)$ and $f_2(\mathcal{D}; \boldsymbol{\theta}_2, \boldsymbol{\theta}_s)$ with shared weights $\boldsymbol{\theta}_s$ and private weights $\boldsymbol{\theta}_1$ and $\boldsymbol{\theta}_2$. We formulate learning as the maximization of the posterior $p(\boldsymbol{\theta}_1, \boldsymbol{\theta}_2, \boldsymbol{\theta}_s | \mathcal{D})$. Under mild assumptions we show that this posterior can be approximated and expressed using a loss, dubbed Weight Plasticity Loss (WPL), that minimizes multi-model forgetting. Our framework evaluates the importance of each weight, conditioned on the previously-trained model, and encourages the update of each shared weight to be inversely proportional to its importance. We then show that our approach extends to more than two models by exploiting it for neural architecture search.

Our work is the first of which we are aware to propose a solution to multi-model forgetting. We establish the merits of our approach when training two models with partially shared weights and in the context of neural architecture search. For the former, we establish the effectiveness of WPL in the strict convergence case, where each model is trained until convergence, and in the more realistic loose convergence setting, where training is stopped early. WPL can reduce the forgetting effect by 99% when model A converges fully, and by 52% in the loose convergence case.

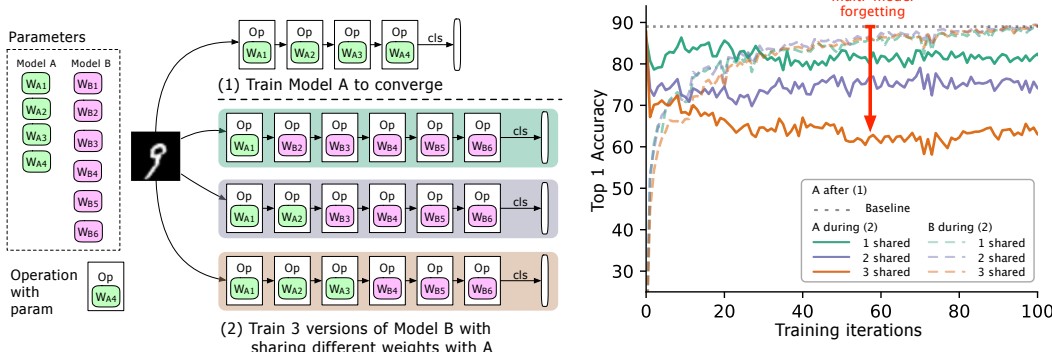

Figure 1: *(Left)* Two models to be trained (A, B), where A's parameters are in green and B's in purple, and B shares some parameters with A (indicated in green during phase 2). We first train A to convergence and then train B. *(Right)* Accuracy of model A as the training of B progresses. The different colors correspond to different numbers of shared layers. The accuracy of A decreases dramatically, especially when more layers are shared, and we refer to the drop (the red arrow) as multi-model forgetting. This experiment was performed on MNIST (LeCun & Cortes, 2010).

For neural architecture search, we implement WPL within the efficient ENAS method of Pham et al. (2018), a state-of-the-art technique that relies on parameter sharing and corresponds to the loose convergence setting. We show that, at each iteration, the use of WPL reduces the forgetting effect by 51% on the most affected model and by 95% on average over all sampled models. Our final results on the best architecture found by the search confirm that limiting multi-model forgetting yields better results and better convergence for both language modeling (on the PTB dataset (Marcus et al., 1994)) and image classification (on the CIFAR10 dataset (Krizhevsky et al., 2009)). For language modeling the perplexity decreases from 65.01 for ENAS without WPL to 61.9 with WPL. For image classification WPL yields a drop of top-1 error from 4.87% to 3.81%. We also adapt our method to NAO (Luo et al., 2018) and show, in appendix due to space limitations, that multi-model forgetting is significantly reduced. We will make our code publicly available upon acceptance of this paper.

## 2 RELATED WORK

**Single-model Forgetting**. The goal of training a single model to tackle multiple problems is to leverage the structures learned for one task for other tasks. This has been employed in transfer learning (Pan & Yang, 2010), multi-task learning (Caruana, 1997) and lifelong learning (Silver et al., 2013). However, sequential learning of later tasks has visible negative consequences for the initial one. Kirkpatrick et al. (2017) selectively slow down the learning of the weights that are comparatively important for the first task by defining the importance of an individual weight using its Fisher information (Rissanen, 1996). He & Jaeger (2017) project the gradient so that directions relevant to the previous task are unaffected. Other families of methods save the older models separately to create progressive networks (Rusu et al., 2016) or use regularization to force the parameters to remain close to the values obtained by previous tasks while learning new ones (Li & Hoiem, 2016). In (Xu & Zhu, 2018), forgetting is avoided altogether by fixing the parameters of the first model while complementing the second one with additional operations found by an architecture search procedure. This work, however, does not address the multi-model forgetting that occurs during the architecture search. An extreme case of sequential learning is lifelong learning, for which the solution to catastrophic forgetting developed by Aljundi et al. (2018) is also to prioritize the weight updates, with smaller updates for weights that are important for previously-learned tasks.

**Parameter Sharing in Neural Architecture Search.** In both sequential learning on multiple tasks and lifelong learning, the forgetfulness concerns an individual model. Here we tackle scenarios where one seeks to optimize a population of multiple models that share parts of their internal structure. The use of multiple models to solve a single task dates back to model ensembles (Dietterich, 2000). Recently, sharing weights between models that are candidate solutions to a problem has shown great promise in the generation of custom neural architectures, known as neural architecture search (Elsken et al., 2018). Existing neural architecture search strategies mostly divide into reinforcement learning and evolutionary techniques. For instance, Zoph & Le (2017) use reinforcement

learning to explore a search space of candidate architectures, with each architecture encoded as a string using an RNN trained with REINFORCE (Williams, 1992) and taking validation performance as the reward. MetaQNN (Baker et al., 2017) uses Q-Learning to design CNN architectures. By contrast, neuro-evolution strategies use evolutionary algorithms (Bäck, 1996) to perform the search. An example is Liu et al. (2018a), who introduce a hierarchical representation of neural networks and use tournament selection (Goldberg & Deb, 1991) to evolve the architectures.

Initial search solutions required hundreds of GPUs due to the huge search space, but recent efforts have made the search more tractable, for example via the use of neural blocks (Negrinho & Gordon, 2017; Bennani-Smires et al., 2018). Similarly, and directly related to this work, *weight sharing* between the candidates has allowed researchers to greatly decrease the computational cost of neural architecture search. For neuroevolution methods, sharing is implicit. For example, Real et al. (2017) define weight inheritance as allowing the children to inherit their parents' weights whenever possible. For RL-base techniques, weight sharing is modeled explicitly and has been shown to lead to significant gains. In particular, ENAS (Pham et al., 2018), which builds upon NAS (Zoph & Le, 2017), represents the search space as a single directed acyclic graph (DAG) in which each candidate architecture is a subgraph. EAS (Cai et al., 2018) also uses an RL strategy to grow the network depth or layer width with function-preserving transformations defined by Chen et al. (2016) where they initialize new models with previous parameters. DARTS (Liu et al., 2018b) uses soft assignment to select paths that implicitly inherit the previous weights. NAO (Luo et al., 2018) replaces the reinforcement learning portion of ENAS with a gradient-based auto-encoder that directly exploits weight sharing. While weight sharing has proven effective, its downsides have never truly been studied. Bender et al. (2018) realized that training was unstable and proposed to circumvent this issue by randomly dropping network paths. However, they did not analyze the reasons underlying the training instability. Here, by contrast, we highlight the underlying multi-model forgetting problem and introduce a statistically-justified solution that further improves on path dropout.

## 3 METHODOLOGY

In this section we study the training of multiple models that share certain parameters. As discussed above, training the multiple models sequentially as in Pham et al. (2018), for example, is suboptimal, since multi-model forgetting arises. Below we derive a method to overcome this for two models, and then show how our formalism extends to multiple models in the context of neural architecture search, and in particular within ENAS (Pham et al., 2018).

### 3.1 WEIGHT PLASTICITY LOSS: PREVENTING MULTI-MODEL FORGETTING

Given a dataset $\mathcal{D}$, we seek to train two architectures $f_1(\mathcal{D}; \boldsymbol{\theta}_1, \boldsymbol{\theta}_s)$ and $f_2(\mathcal{D}; \boldsymbol{\theta}_2, \boldsymbol{\theta}_s)$ with shared parameters $\boldsymbol{\theta}_s$ and private parameters $\boldsymbol{\theta}_1$ and $\boldsymbol{\theta}_2$. We suppose that the models are trained sequentially, which reflects common large-model, large-dataset scenarios and will facilitate generalization. Below, we derive a statistically-motivated framework that prevents multi-model forgetting; it stops the training of the second model from degrading the performance of the first model.

We formulate training as finding the parameters $\boldsymbol{\theta} = (\boldsymbol{\theta}_1, \boldsymbol{\theta}_2, \boldsymbol{\theta}_s)$ that maximize the posterior probability $p(\boldsymbol{\theta} \mid \mathcal{D})$, which we approximate to derive our new loss function. Below we discuss the different steps of this approximation, first expressing $p(\boldsymbol{\theta} \mid \mathcal{D})$ more conveniently.

**Lemma 1.** *Given a dataset $\mathcal{D}$ and two architectures with shared parameters $\boldsymbol{\theta}_s$ and private parameters $\boldsymbol{\theta}_1$ and $\boldsymbol{\theta}_2$, and provided that $p(\boldsymbol{\theta}_1, \boldsymbol{\theta}_2 \mid \boldsymbol{\theta}_s, \mathcal{D}) = p(\boldsymbol{\theta}_1 \mid \boldsymbol{\theta}_s, \mathcal{D})p(\boldsymbol{\theta}_2 \mid \boldsymbol{\theta}_s, \mathcal{D})$, we have*

$$p(\boldsymbol{\theta}_1, \boldsymbol{\theta}_2, \boldsymbol{\theta}_s \mid \mathcal{D}) \propto \frac{p(\mathcal{D} \mid \boldsymbol{\theta}_2, \boldsymbol{\theta}_s)p(\boldsymbol{\theta}_1, \boldsymbol{\theta}_s)p(\boldsymbol{\theta}_2, \boldsymbol{\theta}_s)}{\int p(\mathcal{D} \mid \boldsymbol{\theta}_1, \boldsymbol{\theta}_s)p(\boldsymbol{\theta}_1, \boldsymbol{\theta}_s)d\boldsymbol{\theta}_1}. \tag{1}$$

*Proof.* Provided in the appendix. □

Lemma 1 presupposes that $p(\boldsymbol{\theta}_1, \boldsymbol{\theta}_2 \mid \boldsymbol{\theta}_s, \mathcal{D}) = p(\boldsymbol{\theta}_1 \mid \boldsymbol{\theta}_s, \mathcal{D})p(\boldsymbol{\theta}_2 \mid \boldsymbol{\theta}_s, \mathcal{D})$, i.e., $\boldsymbol{\theta}_1$ and $\boldsymbol{\theta}_2$ are conditionally independent given $\boldsymbol{\theta}_s$ and the dataset $\mathcal{D}$. While this must be checked in applications, it is suitable for our setting, since we want both networks, $f_1(\mathcal{D}; \boldsymbol{\theta}_1, \boldsymbol{\theta}_s)$ and $f_2(\mathcal{D}; \boldsymbol{\theta}_2, \boldsymbol{\theta}_s)$, to train independently well.

To derive our loss we study the components on the right of equation (1). We start with the integral in the denominator, for which we seek a closed form. Suppose we have trained the first model and seek to update the parameters of the second one while avoiding forgetting. The following lemma provides an expression for the denominator of equation (1).

**Lemma 2.** *Suppose we have the maximum likelihood estimate $(\hat{\boldsymbol{\theta}}_1, \hat{\boldsymbol{\theta}}_s)$ for the first model, write* $\mathrm{Card}(\boldsymbol{\theta}_1) + \mathrm{Card}(\boldsymbol{\theta}_s) = p_1 + p_s = p$, *and let the negative Hessian $\boldsymbol{H}_p(\hat{\boldsymbol{\theta}}_1, \hat{\boldsymbol{\theta}}_s)$ of the log poste-rior probability distribution $\log p(\boldsymbol{\theta}_1, \boldsymbol{\theta}_s \mid \mathcal{D})$ evaluated at $(\hat{\boldsymbol{\theta}}_1, \hat{\boldsymbol{\theta}}_s)$ be partitioned into four blocks corresponding to $(\boldsymbol{\theta}_1, \boldsymbol{\theta}_s)$ as*

$$\boldsymbol{H}_p(\hat{\boldsymbol{\theta}}_1, \hat{\boldsymbol{\theta}}_s) = \left[ \begin{array}{c|c} \boldsymbol{H}_{11} & \boldsymbol{H}_{1s} \\ \hline \boldsymbol{H}_{s1} & \boldsymbol{H}_{ss} \end{array} \right].$$

*If the parameters of each model follow Normal distributions, i.e., $(\boldsymbol{\theta}_1, \boldsymbol{\theta}_s) \sim \mathcal{N}_p(\boldsymbol{0}, \sigma^2 \boldsymbol{I}_p)$, with $\boldsymbol{I}_p$ the $p$-dimensional identity matrix, then*

$$\int p(\mathcal{D} \mid \boldsymbol{\theta}_1, \boldsymbol{\theta}_s) p(\boldsymbol{\theta}_s, \boldsymbol{\theta}_1) d\boldsymbol{\theta}_1 = \exp\{l_p(\hat{\boldsymbol{\theta}}_1, \hat{\boldsymbol{\theta}}_s) - \frac{1}{2}\boldsymbol{v}^\top \boldsymbol{\Omega} \boldsymbol{v}\} \times (2\pi)^{p_1/2} |\det(\boldsymbol{H}_{11}^{-1})|^{1/2}, \quad (2)$$

*where $\boldsymbol{v} = \boldsymbol{\theta}_s - \hat{\boldsymbol{\theta}}_s$ and $\boldsymbol{\Omega} = \boldsymbol{H}_{ss} - \boldsymbol{H}_{1s}^\top \boldsymbol{H}_{11}^{-1} \boldsymbol{H}_{1s}$ .*

*Proof.* Provided in the appendix. $\qquad\square$

Lemma 2 requires the maximum likelihood estimate $(\hat{\boldsymbol{\theta}}_1, \hat{\boldsymbol{\theta}}_s)$, which can be hard to obtain with deep networks, since they have non-convex objective functions. In practice, one can train the network to convergence and treat the resulting parameters as maximum likelihood estimates. Our experiments show that the parameters obtained without optimizing to convergence can be used effectively. More-over Haeffele & Vidal (2017) showed that networks relying on *positively homogeneous functions* have critical points that are either global minimizers or saddle points, and that training to conver-gence yields near-optimal solutions, which correspond to true maximum likelihood estimates.

Following Lemmas 1 and 2, as shown in the appendix,

$$\log p(\boldsymbol{\theta} \mid \mathcal{D}) \propto \log p(\mathcal{D} \mid \boldsymbol{\theta}_2, \boldsymbol{\theta}_s) + \log p(\boldsymbol{\theta}_2, \boldsymbol{\theta}_s) + \log p(\boldsymbol{\theta}_1, \boldsymbol{\theta}_s \mid \mathcal{D}) + \frac{1}{2}\boldsymbol{v}^\top \boldsymbol{\Omega} \boldsymbol{v}. \quad (3)$$

To derive a loss function that prevents multi-model forgetting, consider equation (3). The first term on its right-hand side corresponds to the log likelihood of the second model and can be replaced by the cross-entropy $\mathcal{L}_2(\boldsymbol{\theta}_2, \boldsymbol{\theta}_s)$, and if we use a Gaussian prior on the parameters, the second term encodes an $L^2$ regularization. Since equation (3) depends only on the log likelihood of the second model $f_2(\mathcal{D}; \boldsymbol{\theta}_2, \boldsymbol{\theta}_s)$, the information learned from the first model $f_1(\mathcal{D}; \boldsymbol{\theta}_1, \boldsymbol{\theta}_s)$ must reside in the conditional posterior probability $\log p(\boldsymbol{\theta}_1, \boldsymbol{\theta}_s \mid \mathcal{D})$, and the final term, $\frac{1}{2}\boldsymbol{v}^\top \boldsymbol{\Omega} \boldsymbol{v}$, must represent the interactions between the models $f_1(\mathcal{D}; \boldsymbol{\theta}_2, \boldsymbol{\theta}_s)$ and $f_2(\mathcal{D}; \boldsymbol{\theta}_1, \boldsymbol{\theta}_s)$. This term will not appear in a standard single-model forgetting scenario. Let us examine these terms more closely.

The posterior probability $p(\boldsymbol{\theta}_1, \boldsymbol{\theta}_s \mid \mathcal{D})$ is intractable, so we apply a Laplace approximation (MacKay, 1992); we approximate the log posterior using a second-order Taylor expansion around the maximum likelihood estimate $(\hat{\boldsymbol{\theta}}_1, \hat{\boldsymbol{\theta}}_s)$. This yields

$$\log p(\boldsymbol{\theta}_1, \boldsymbol{\theta}_s \mid \mathcal{D}) = \log p(\hat{\boldsymbol{\theta}}_1, \hat{\boldsymbol{\theta}}_s \mid \mathcal{D}) - \frac{1}{2}[(\boldsymbol{\theta}_1, \boldsymbol{\theta}_s) - (\hat{\boldsymbol{\theta}}_1, \hat{\boldsymbol{\theta}}_s)]^\top \boldsymbol{H}_p[(\boldsymbol{\theta}_1, \boldsymbol{\theta}_s) - (\hat{\boldsymbol{\theta}}_1, \hat{\boldsymbol{\theta}}_s)], \quad (4)$$

where $\boldsymbol{H}_p(\hat{\boldsymbol{\theta}}_1, \hat{\boldsymbol{\theta}}_s)$ is the negative Hessian of the log posterior evaluated at the maximum likelihood estimate. As the first derivative is evaluated at the maximum likelihood estimate, it equals zero.

Equation (4) yields a Gaussian approximation to the posterior with mean $(\hat{\boldsymbol{\theta}}_1, \hat{\boldsymbol{\theta}}_s)$ and covariance matrix $\boldsymbol{H}_p^{-1}$, i.e.,

$$p(\boldsymbol{\theta}_1, \boldsymbol{\theta}_s \mid \mathcal{D}) \propto \exp\left\{ -\frac{1}{2}[(\boldsymbol{\theta}_1, \boldsymbol{\theta}_s) - (\hat{\boldsymbol{\theta}}_1, \hat{\boldsymbol{\theta}}_s)]^\top \boldsymbol{H}_p[(\boldsymbol{\theta}_1, \boldsymbol{\theta}_s) - (\hat{\boldsymbol{\theta}}_1, \hat{\boldsymbol{\theta}}_s)] \right\}. \quad (5)$$

Our parameter space is too large to compute the inverse of the negative Hessian $\boldsymbol{H}_p$, so we replace it with the diagonal of the Fisher information, $\mathrm{diag}(\boldsymbol{F})$. This approximation falsely presupposes that

the parameters $(\boldsymbol{\theta}_1, \boldsymbol{\theta}_s)$ are independent, but it has already proven effective (Kirkpatrick et al., 2017; Pascanu & Bengio, 2014). One of its main advantages is that we can compute the Fisher information from the squared gradients, thereby avoiding any need for second derivatives.

Using equation (5) and the Fisher approximation we can express the log posterior as

$$\log p(\boldsymbol{\theta}_1, \boldsymbol{\theta}_s \mid \mathcal{D}) \propto \frac{\alpha}{2} \sum_{\theta_{s_i} \in \boldsymbol{\theta}_s} F_{\theta_{s_i}} (\theta_{s_i} - \hat{\theta}_{s_i})^2 \,, \tag{6}$$

where $F_{\theta_{s_i}}$ is the diagonal element corresponding to parameter $\theta_{s_i}$ in the diagonal approximation of the Fisher information matrix, which can be obtained from the trained model $f_1(\mathcal{D}; \boldsymbol{\theta}_1, \boldsymbol{\theta}_s)$.

Now consider the last term in equation (3), noting that $\boldsymbol{\Omega} = \boldsymbol{H}_{ss} - \boldsymbol{H}_{1s}^{\top} \boldsymbol{H}_{11}^{-1} \boldsymbol{H}_{1s}$, as defined in Lemma 2. As our previous approximation relies on the assumption of a diagonal Fisher information matrix, we have $\boldsymbol{H}_{1s} = \boldsymbol{0}$, leading to $\boldsymbol{\Omega} = \boldsymbol{H}_{ss}$, so

$$\frac{1}{2} \boldsymbol{v}^{\top} \boldsymbol{\Omega} \boldsymbol{v} = \frac{1}{2} \sum_{\theta_{s_i} \in \boldsymbol{\theta}_s} F_{\theta_{s_i}} (\theta_{s_i} - \hat{\theta}_{s_i})^2 \,. \tag{7}$$

The last two terms on the right-hand side of equation (3), as expressed in equation (6) and equation (7), can then be grouped. Combining the result with the first two terms, discussed below equation (3), yields our Weight Plasticity Loss,

$$\mathcal{L}_{\text{WPL}}(\boldsymbol{\theta}_2, \boldsymbol{\theta}_s) = \mathcal{L}_2(\boldsymbol{\theta}_2, \boldsymbol{\theta}_s) + \frac{\lambda}{2}(\|\boldsymbol{\theta}_s\|^2 + \|\boldsymbol{\theta}_2\|^2) + \frac{\alpha}{2} \sum_{\theta_{s_i} \in \boldsymbol{\theta}_s} F_{\theta_{s_i}} (\theta_{s_i} - \hat{\theta}_{s_i})^2, \tag{8}$$

where $F_{\theta_{s_i}}$ is the diagonal element corresponding to parameter $\theta_{s_i}$ in the Fisher information matrix obtained from the trained first model $f_1(\mathcal{D}; \boldsymbol{\theta}_1, \boldsymbol{\theta}_s)$. We omit the terms depending on $\boldsymbol{\theta}_1$ in equation (6) because we are optimizing with respect to $(\boldsymbol{\theta}_2, \boldsymbol{\theta}_s)$ at this stage. The Fisher information in the last term encodes the importance of each shared weight for the first model's performance, so WPL encourages preserving any shared parameters that were important for the first model, while allowing others to undergo larger changes and thus to improve the accuracy of the second model.

### 3.1.1 RELATION TO ELASTIC WEIGHT CONSOLIDATION

The final loss function obtained in equation (8) may appear similar to that obtained by Kirkpatrick et al. (2017) when formulating their Elastic Weight Consolidation (EWC) to address catastrophic forgetting. However, the problem we address here is fundamentally different. Kirkpatrick et al. (2017) tackle sequential learning on different tasks, where a *single model* is sequentially trained using *two datasets*, and their goal is to maximize the posterior $p(\boldsymbol{\theta} \mid \mathcal{D}) = p(\boldsymbol{\theta} \mid \mathcal{D}_1, \mathcal{D}_2)$. By relying on Laplace approximations in neural networks (MacKay, 1992) and the connection between the Fisher information matrix and second-order derivatives (Pascanu & Bengio, 2014), EWC is then formulated as the loss $\mathcal{L}(\boldsymbol{\theta}) = \mathcal{L}_{\text{B}}(\boldsymbol{\theta}) + \sum_i \frac{\lambda}{2} F_i (\theta_i - \theta_{\text{A},i}^{\star})^2$, where A and B refer to *two different tasks*, $\boldsymbol{\theta}$ encodes the network parameters and $F_i$ is the Fisher information of $\theta_i$.

Here we consider scenarios with a *single dataset* but *two models* with shared parameters, and aim to maximize the posterior $p(\boldsymbol{\theta}_1, \boldsymbol{\theta}_2, \boldsymbol{\theta}_s \mid \mathcal{D})$. The resulting WPL combines the original loss of the second model, a Fisher-weighted MSE term on the shared parameters and an $L^2$ regularizer on the parameters of the second model. More importantly, the last term in equation (3), $\boldsymbol{v}^{\top} \boldsymbol{\Omega} \boldsymbol{v}$, is specific to the multi-model case, since it encodes the interaction between the two models; it never appears in the EWC derivation. Because we adopt a Laplace approximation based on the diagonal Fisher information matrix, as shown in Equation (7), this term can then be grouped with that of Equation (6), making our WPL loss appear similar to the EWC loss. In principle, however, other approximations of $\boldsymbol{v}^{\top} \boldsymbol{\Omega} \boldsymbol{v}$ could be used, such as a Laplace one with a full covariance matrix, which would yield a final loss that differs fundamentally from the EWC one. In any event, under mild assumptions we obtain a statistically-motivated loss function that is useful in practice. We believe this to be a valuable contribution in itself, but, more importantly, we show below that it can significantly reduce multi-model forgetting.

### 3.2 WPL FOR NEURAL ARCHITECTURE SEARCH

In the previous section, we considered only two models being trained sequentially, but in practice one often seeks to train three or more models. Our approach is then unchanged, but each model

shares parameters with several other models, which entails using diagonal approximations to Fisher information matrices for all previously-trained models from equation (3). In the remainder of this section, we discuss how our approach can be used for neural architecture search.

Consider using our WPL within the ENAS strategy of Pham et al. (2018). ENAS is a reinforcement-learning-based method that consists of two training processes: 1) sequentially train sampled models with shared parameters; and 2) train a controller RNN that generates model candidates. Incorporating our WPL within ENAS only affects 1).

The first step of ENAS consists of sampling a fixed number of architectures from the RNN controller, and training each architecture on $B$ batches. This implies that our requirement for access to the maximum likelihood estimate of the previously-trained models is not satisfied, but we verify that in practice our WPL remains effective in this scenario. After sufficiently many epochs it is likely that all the parameters of a newly-sampled architecture are shared with previously-trained ones, and then we can consider that all parameters of new models are shared.

At the beginning of the search, the parameters of all models are randomly initialized. Adopting WPL directly from the start would therefore make it hard for the process to learn anything, as it would encourage some parameters to remain random. To better satisfy our assumption that the parameters of previously-trained models should be optimal, we follow the original ENAS training strategy for $n$ epochs, with $n = 5$ for RNN search and $n = 3$ for CNN search in our experiments. We then incorporate our WPL and store the optimal parameters after each architecture is trained. We also update the Fisher information, which adds virtually no computational overhead, because $F_{\theta i} = (\partial \mathcal{L}/\partial \theta_i)^2$, where $\mathcal{L} = \sum_i \mathcal{L}_i$, with $i$ indexing the previously-sampled architectures, and the derivatives are already computed for back-propagation. To ensure that these updates use the contributions from all previously-sampled architectures, we use a momentum-based update expressed as $F_{\theta_i}^t = (1 - \eta)F_{\theta_i}^{t-1} + \eta(\partial \mathcal{L}/\partial \theta_i)^2$, with $\eta = 0.9$. Since such Fisher information is not computed at the MLE of the parameters, we flush the global Fisher buffer to zero every three epochs, yielding an increasingly accurate estimate of the Fisher information as optimization proceeds. We also use a scheduled decay for $\alpha$ in equation (8).

## 4 EXPERIMENTS

We first evaluate our weight plasticity loss (WPL) in the general scenario of training two models sequentially, both in the strict convergence case and when the weights of the first model are sub-optimal. We then evaluate the performance of our approach within the ENAS framework.

### 4.1 GENERAL SCENARIO: TRAINING TWO MODELS

To test WPL in the general scenario, we used the MNIST handwritten digit recognition dataset (LeCun & Cortes, 2010). We designed two feed-forward networks with 4 (Model A) and 6 (Model B) layers, respectively. All the layers of A are shared by B.

Let us first evaluate our approach in the strict convergence case. To this end, we trained A until convergence, thus obtaining a solution close to the MLE $\hat{\boldsymbol{\theta}}_A = (\hat{\boldsymbol{\theta}}_1, \hat{\boldsymbol{\theta}}_s)$, since all our operations are positively homogeneous (Haeffele & Vidal, 2017). To compute the Fisher information, we used the backward gradients of $\boldsymbol{\theta}_s$ calculated on 200 images in the validation set. We then initialized $\boldsymbol{\theta}_s$ of Model B, $f_B(\mathcal{D}; (\boldsymbol{\theta}_2, \boldsymbol{\theta}_s))$, as $\hat{\boldsymbol{\theta}}_s$ and trained B by standard SGD with respect to all its parameters. Figure 2(a) compares the performance of training model B with and without WPL. Without WPL the performance of A degrades as training B progresses, but using WPL allows us to maintain the initial performance of A, indicated as Baseline in the plot. This entails no loss of performance for B, whose final accuracy is virtually the same both with and without WPL.

The assumption of optimal weights is usually hard to enforce. We therefore now turn to the more realistic loose convergence scenario. To evaluate the influence of sub-optimal weights for Model A on our approach, we trained Model A to different, increasingly lower, top 1 accuracies. As shown in Figure 2(b) and (c), even in this setting our approach still significantly reduces multi-model forgetting. We can quantify the relative reduction rate of such forgetting as $d_A - d_{A+WPL}/d_A$, where $d = acc_A^* - acc$ is A's accuracy decay after training B. Our WPL can reduce multi-model forgetting. up to 99% for more converged model, and by 52% even for the loose case. This suggests

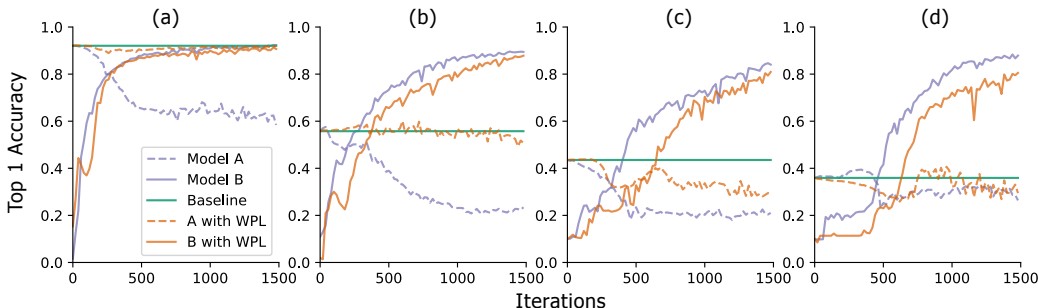

Figure 2: **From strict to loose convergence.** We conduct experiments on MNIST with models A and B with shared parameters, and report the accuracy of Model A before training Model B (baseline, green) and the accuracy of Models A and B while training Model B with (orange) or without (blue) WPL. In *(a)* we show the results for strict convergence: A is initially trained to convergence. We then relax this assumption and train A to around 55% *(b)*, 43% *(c)*, and 38% *(d)* of its optimal accuracy. We see that WPL is highly effective when A is trained to at least 40% of optimality; below, the Fisher information becomes too inaccurate to provide reliable importance weights. Thus WPL helps to reduce multi-model forgetting, even when the weights are not optimal. WPL reduced forgetting by up to 99.99% for *(a)* and *(b)*, and by up to 2% for *(c)*.

that the Fisher information remains a reasonable empirical approximation to the weights' importance even when our optimality assumption is not satisfied.

## 4.2   WPL FOR NEURAL ARCHITECTURE SEARCH

We demonstrate the effectiveness of WPL in a real-world application, neural architecture search. We incorporate WPL in the ENAS framework (Pham et al., 2018), which relies on weight-sharing across model candidates to speed up the search and thus, while effective, will suffer from multi-model forgetting even with random dropping weights and output dropout. To show this, we examine how the previously-trained architectures are affected by the training of new ones by evaluating the prediction error of each sampled architecture on a fraction of the validation dataset immediately after it is trained, denoted by $err_1$, and at the end of the epoch, denoted by $err_2$. A positive difference $err_2 - err_1$ for a specific architecture indicates that it has been forced to forget by others.

We performed two experiments: RNN cell search on the PTB dataset and CNN micro-cell search on the CIFAR10 dataset. We report the mean error difference for all sampled architectures, the mean error difference for the 5 architectures with the lowest $err_1$, and the maximum error difference over all sampled architectures. Figure 3*(a)*, *(b)* and *(c)* plot these as functions of the training epochs for the RNN case, and similar plots for CNN search are in the appendix. The plots shows that without WPL the error differences are much larger than 0, clearly displaying the multi-model forgetting effect. This is particularly pronounced in the first half of training, which can have a dramatic effect on the final results, as it corresponds to the phase where the algorithm searches for promising architectures. WPL significantly reduces the forgetting, as shown by much lower error differences. With WPL, these differences tend to decrease over time, emphasizing that the observed Fisher information encodes an increasingly reliable notion of weight importance as training progresses. Owing to limited computational resources we estimate the Fisher information using only small validation batches, but use of larger batches could further improve our results.

In Figure 3*(d)*, we plot the average reward of all sampled architectures as a function of the training iterations. In the first half of training, the models trained with WPL tend to have lower rewards. This can be explained by the use of a large value for $\alpha$ in equation (8) during this phase; while such a large value may prevent the best models from achieving as high a reward as possible, it has the advantage of preventing the forgetting of good models, and thus avoiding their being discarded early. This is shown by the fact that, in the second half of training, when we reduce $\alpha$, the mean reward of the architectures trained with WPL is higher than without using it. In other words, our approach allows us to maintain better models until the end of training.

Table 1: **Results of the best models found.** We take the best model obtained during the search and train it from scratch. ENAS* corresponds to the results of Pham et al. (2018) obtained after extensive hyper-parameter search, while ENAS and ENAS+WPL were trained in comparable conditions. For both RNN and CNN search, our WPL gives a significant boost to ENAS, thus showing the importance of overcoming multi-model forgetting. In the RNN case, our approach outperforms ENAS* without requiring extensive hyper-parameter tuning. The best results in each row are bold.

| Datasets | Metric | ENAS* | ENAS | **ENAS + WPL** |
|----------|--------|-------|------|----------------|
| PTB | perplexity | 63.26 | 65.01 | **61.9** |
| CIFAR10 | top-1 error | **3.54** | 4.87 | 3.81 |

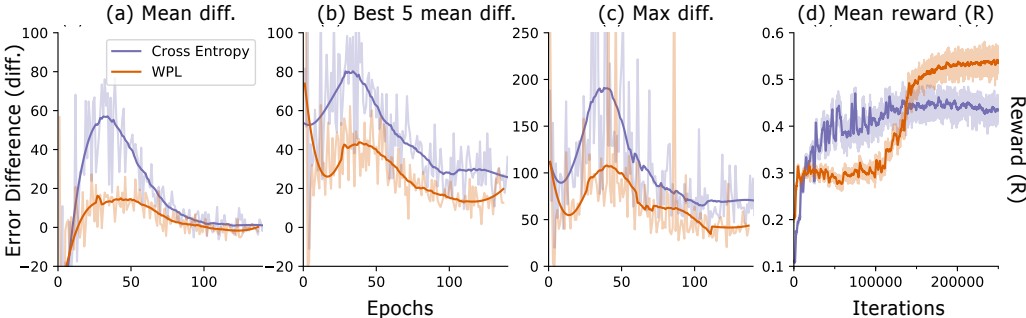

Figure 3: **Error difference during neural architecture search.** For each architecture, we compute the RNN error differences $err_2 - err_1$, where $err_1$ is the error right after training this architecture and $err_2$ the one after all architectures are trained in the current epoch. We plot (a) the mean difference over all sampled models, (b) the mean difference over the 5 models with lowest $err_1$, and (c) the max difference over all models. The plots show that WPL reduces multi-model forgetting; the error differences are much closer to 0. Quantitatively, the forgetting reduction can be up to 95% for *(a)*, 59% for *(b)* and 51% for *(c)*. In (d), we plot the average reward of the sampled architectures as a function of training iterations. Although WPL initially leads to lower rewards, due to a large weight $\alpha$ in equation (8), by reducing the forgetting it later allows the controller to sample better architectures, as indicated by the higher reward in the second half.

When the search is over, we train the best architecture from scratch and evaluate its final accuracy. Table 1 compares the results obtained without (ENAS) and with WPL (ENAS+WPL) with those from the original ENAS paper (ENAS*), which were obtained after conducting an extensive hyper-parameter search. For both datasets, using WPL improves final model accuracy, thus showing the importance of overcoming multi-model forgetting. In the case of PTB, our approach even outperforms ENAS*, without extensive hyper-parameter tuning. Based on the gap between ENAS and ENAS*, we anticipate that such a tuning procedure could further boost our results. In any event, we believe that these results already clearly show the benefits of reducing multi-model forgetting.

## 5 CONCLUSION

This paper has identified the problem of multi-model forgetting in the context of sequentially training multiple models: the shared weights of previously-trained models are overwritten during training of subsequent models, leading to performance degradation. We show that the degree of degradation is linked to the proportion of shared weights, and introduce a statistically-motivated weight plasticity loss (WPL) to overcome this. Our experiments on multi-model training and on neural architecture search clearly show the effectiveness of WPL in reducing multi-model forgetting and yielding better architectures, leading to improved results in both natural language processing and computer vision tasks. We believe that the impact of WPL goes beyond the tasks studied in this paper. In future work, we plan to integrate WPL within other neural architecture search strategies in which weight sharing occurs and to study its use in other multi-model contexts, such as for ensemble learning.

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

# A SUPPLEMENTARY EXPERIMENTS

## A.1 NEURAL ARCHITECTURE OPTIMIZATION

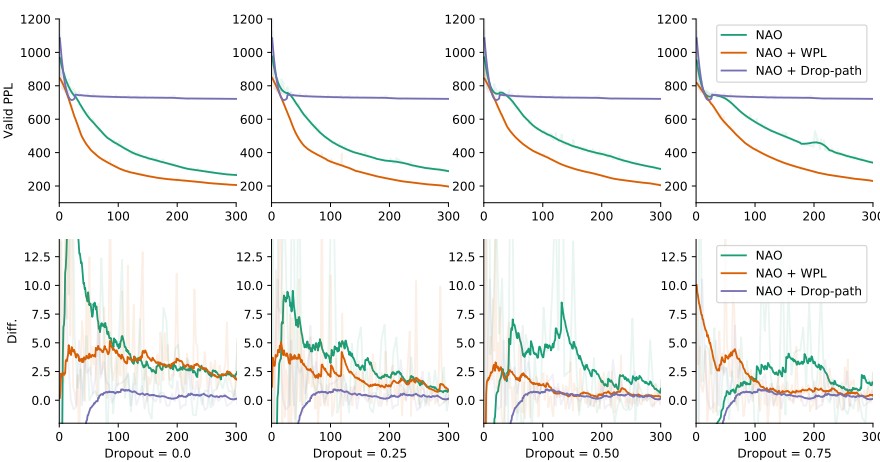

Figure 4: **Comparison of different output dropout rates for NAO.** We plot the mean validation perplexity while searching the best architecture (top) and the best 5 model's error differences (bottom) for four different dropout rates. Note that path dropping in NAO prevents learning shortly after model initialization. At all the dropout rates, our WPL achieves lower error differences, i.e., it reduces multi-model forgetting, as well as speeds up training.

Our approach is general, and its use in the context of neural architecture search is not limited to ENAS. To demonstrate this, we applied it to the neural architecture optimization (NAO) method of Luo et al. (2018), which also exploits weight-sharing in the search phase. In this context, we therefore investigate (i) whether multi-model forgetting occurs, and if so, (ii) the effectiveness of our approach in the NAO framework. Due to resource and time constraints, we focus our experiments mainly on the search phase, as training the best searched model from scratch takes around 4 GPU days. To evaluate the influence of the dropout strategy of Bender et al. (2018), we test NAO with or without random path-dropping and with four output dropout rates from 0 to 0.75 by steps of 0.25. As in Section 4.2, in Figure 4, we plot the mean validation perplexity and the best five model's error differences for all models that are sampled during a single training epoch. For random path-dropping, since Luo et al. (2018) exploit a more aggressive dropping policy than that used in (Bender et al., 2018), we can see that validation perplexity quickly plateaus. Hence we do not add our WPL to the path dropout strategy, but use it in conjunction with output dropout.

At all four different dropout rates, WPL clearly reduces multi-model forgetting and accelerates training. The level of forgetting decreases with the dropout rate, but our loss always further reduces it. Among the three methods, Nao + path dropping suffers the least from forgetting. However, this is only due to the fact that it does not learn properly. By contrast, our WPL reduces multi-model forgetting while still allowing the models to learn. This shows that our approach generalizes beyond ENAS for neural architecture search.

# B PROOFS

**Lemma 1.** *Given a dataset $\mathcal{D}$ and two architectures with shared parameters $\boldsymbol{\theta}_s$ and private parameters $\boldsymbol{\theta}_1$ and $\boldsymbol{\theta}_2$, and provided that $p(\boldsymbol{\theta}_1, \boldsymbol{\theta}_2 \mid \boldsymbol{\theta}_s, \mathcal{D}) = p(\boldsymbol{\theta}_1 \mid \boldsymbol{\theta}_s, \mathcal{D})p(\boldsymbol{\theta}_2 \mid \boldsymbol{\theta}_s, \mathcal{D})$, we have*

$$p(\boldsymbol{\theta}_1, \boldsymbol{\theta}_2, \boldsymbol{\theta}_s \mid \mathcal{D}) \propto \frac{p(\mathcal{D} \mid \boldsymbol{\theta}_2, \boldsymbol{\theta}_s)p(\boldsymbol{\theta}_1, \boldsymbol{\theta}_s)p(\boldsymbol{\theta}_2, \boldsymbol{\theta}_s)}{\int p(\mathcal{D} \mid \boldsymbol{\theta}_1, \boldsymbol{\theta}_s)p(\boldsymbol{\theta}_1, \boldsymbol{\theta}_s)d\boldsymbol{\theta}_1}. \tag{1}$$

*Proof.* Using Bayes' theorem and ignoring constants, we have

$$p(\boldsymbol{\theta} \mid \mathcal{D}) = \frac{p(\boldsymbol{\theta}_1, \boldsymbol{\theta}_2, \boldsymbol{\theta}_s, \mathcal{D})}{p(\mathcal{D})}$$

$$\propto p(\boldsymbol{\theta}_1 \mid \boldsymbol{\theta}_2, \boldsymbol{\theta}_s, \mathcal{D})p(\boldsymbol{\theta}_2, \boldsymbol{\theta}_s, \mathcal{D})$$

$$= p(\boldsymbol{\theta}_1 \mid \boldsymbol{\theta}_s, \mathcal{D})p(\mathcal{D} \mid \boldsymbol{\theta}_2, \boldsymbol{\theta}_s)p(\boldsymbol{\theta}_2, \boldsymbol{\theta}_s)$$

$$\propto \frac{p(\boldsymbol{\theta}_1, \boldsymbol{\theta}_s, \mathcal{D})p(\mathcal{D} \mid \boldsymbol{\theta}_2, \boldsymbol{\theta}_s)p(\boldsymbol{\theta}_2, \boldsymbol{\theta}_s)}{p(\mathcal{D}, \boldsymbol{\theta}_s)}$$

$$\propto \frac{p(\boldsymbol{\theta}_1, \boldsymbol{\theta}_s, \mathcal{D})p(\mathcal{D} \mid \boldsymbol{\theta}_2, \boldsymbol{\theta}_s)p(\boldsymbol{\theta}_2, \boldsymbol{\theta}_s)}{\int p(\mathcal{D} \mid \boldsymbol{\theta}_1, \boldsymbol{\theta}_s)p(\boldsymbol{\theta}_s, \boldsymbol{\theta}_1)d\boldsymbol{\theta}_1}$$

$$\propto \frac{p(\boldsymbol{\theta}_1, \boldsymbol{\theta}_s \mid \mathcal{D})p(\mathcal{D} \mid \boldsymbol{\theta}_2, \boldsymbol{\theta}_s)p(\boldsymbol{\theta}_2, \boldsymbol{\theta}_s)}{\int p(\mathcal{D} \mid \boldsymbol{\theta}_1, \boldsymbol{\theta}_s)p(\boldsymbol{\theta}_s, \boldsymbol{\theta}_1)d\boldsymbol{\theta}_1},$$

where we used the conditional independence assumption $p(\boldsymbol{\theta}_1 \mid \boldsymbol{\theta}_2, \boldsymbol{\theta}_s, \mathcal{D}) = p(\boldsymbol{\theta}_1 \mid \boldsymbol{\theta}_s, \mathcal{D})$ in the third line. $\square$

We now derive a closed-form expression for the denominator of equation (1).

**Lemma 2.** *Suppose we have the maximum likelihood estimate* $(\hat{\boldsymbol{\theta}}_1, \hat{\boldsymbol{\theta}}_s)$ *for the first model, write* $\mathrm{Card}(\boldsymbol{\theta}_1) + \mathrm{Card}(\boldsymbol{\theta}_s) = p_1 + p_s = p$, *and let the negative Hessian* $\boldsymbol{H}_p(\hat{\boldsymbol{\theta}}_1, \hat{\boldsymbol{\theta}}_s)$ *of the log posterior probability distribution* $\log p(\boldsymbol{\theta}_1, \boldsymbol{\theta}_s \mid \mathcal{D})$ *evaluated at* $(\hat{\boldsymbol{\theta}}_1, \hat{\boldsymbol{\theta}}_s)$ *be partitioned into four blocks corresponding to* $(\boldsymbol{\theta}_1, \boldsymbol{\theta}_s)$ *as*

$$\boldsymbol{H}_p(\hat{\boldsymbol{\theta}}_1, \hat{\boldsymbol{\theta}}_s) = \left[ \begin{array}{c|c} \boldsymbol{H}_{11} & \boldsymbol{H}_{1s} \\ \hline \boldsymbol{H}_{s1} & \boldsymbol{H}_{ss} \end{array} \right].$$

*If the parameters of each model follow Normal distributions, i.e.,* $(\boldsymbol{\theta}_1, \boldsymbol{\theta}_s) \sim \mathcal{N}_p(\boldsymbol{0}, \sigma^2 \boldsymbol{I}_p)$, *with* $\boldsymbol{I}_p$ *the* $p$-*dimensional identity matrix, then*

$$\int p(\mathcal{D} \mid \boldsymbol{\theta}_1, \boldsymbol{\theta}_s)p(\boldsymbol{\theta}_s, \boldsymbol{\theta}_1)d\boldsymbol{\theta}_1 = \exp\left\{l_p(\hat{\boldsymbol{\theta}}_1, \hat{\boldsymbol{\theta}}_s) - \frac{1}{2}\boldsymbol{v}^\top \boldsymbol{\Omega}\boldsymbol{v}\right\} \times (2\pi)^{p_1/2}|\det(\boldsymbol{H}_{11}^{-1})|^{1/2}, \quad (2)$$

*where* $\boldsymbol{v} = \boldsymbol{\theta}_s - \hat{\boldsymbol{\theta}}_s$ *and* $\boldsymbol{\Omega} = \boldsymbol{H}_{ss} - \boldsymbol{H}_{1s}^\top \boldsymbol{H}_{11}^{-1}\boldsymbol{H}_{1s}$.

*Proof.* We have

$$p(\mathcal{D} \mid \boldsymbol{\theta}_1, \boldsymbol{\theta}_s)p(\boldsymbol{\theta}_s, \boldsymbol{\theta}_1) \propto e^{l(\boldsymbol{\theta}_1, \boldsymbol{\theta}_s) - (\boldsymbol{\theta}_1, \boldsymbol{\theta}_s)^T(\boldsymbol{\theta}_1, \boldsymbol{\theta}_s)/2\sigma^2} = e^{l_p(\boldsymbol{\theta}_1, \boldsymbol{\theta}_s)},$$

where $l(\boldsymbol{\theta}_1, \boldsymbol{\theta}_s) = \log p(\mathcal{D} \mid \boldsymbol{\theta}_1, \boldsymbol{\theta}_s)$, and $l_p(\boldsymbol{\theta}_1, \boldsymbol{\theta}_s) = l(\boldsymbol{\theta}_1, \boldsymbol{\theta}_s) - (\boldsymbol{\theta}_1, \boldsymbol{\theta}_s)^T(\boldsymbol{\theta}_1, \boldsymbol{\theta}_s)/2\sigma^2$.

Let $\boldsymbol{H}_p(\boldsymbol{\theta}_1, \boldsymbol{\theta}_s) = \boldsymbol{H}(\boldsymbol{\theta}_1, \boldsymbol{\theta}_s) + \sigma^{-2}\boldsymbol{I}_p$ be the negative Hessian of $l_p(\boldsymbol{\theta}_1, \boldsymbol{\theta}_s)$, with $\boldsymbol{I}_p$ the $p$-dimensional identity matrix and $\boldsymbol{H}(\boldsymbol{\theta}_1, \boldsymbol{\theta}_s)$ the negative Hessian of $l(\boldsymbol{\theta}_1, \boldsymbol{\theta}_s)$.

Using the second-order Taylor expansion of $l_p(\boldsymbol{\theta}_1, \boldsymbol{\theta}_s)$ around its maximum likelihood estimate $(\hat{\boldsymbol{\theta}}_1, \hat{\boldsymbol{\theta}}_s)$, we have

$$l_p(\boldsymbol{\theta}_1, \boldsymbol{\theta}_s) = l_p(\hat{\boldsymbol{\theta}}_1, \hat{\boldsymbol{\theta}}_s) - \frac{1}{2}[(\boldsymbol{\theta}_1, \boldsymbol{\theta}_s) - (\hat{\boldsymbol{\theta}}_1, \hat{\boldsymbol{\theta}}_s)]^T \boldsymbol{H}_p(\hat{\boldsymbol{\theta}}_1, \hat{\boldsymbol{\theta}}_s)[(\boldsymbol{\theta}_1, \boldsymbol{\theta}_s) - (\hat{\boldsymbol{\theta}}_1, \hat{\boldsymbol{\theta}}_s)]; \quad (9)$$

the first derivative is zero since it is evaluated at the maximum likelihood estimate. We now partition our negative Hessian matrix as

$$\boldsymbol{H}_p(\hat{\boldsymbol{\theta}}_1, \hat{\boldsymbol{\theta}}_s) = \left[ \begin{array}{c|c} \boldsymbol{H}_{11} & \boldsymbol{H}_{1s} \\ \hline \boldsymbol{H}_{s1} & \boldsymbol{H}_{ss} \end{array} \right],$$

which gives

$$\boldsymbol{A} = [(\boldsymbol{\theta}_1, \boldsymbol{\theta}_s) - (\hat{\boldsymbol{\theta}}_1, \hat{\boldsymbol{\theta}}_s)]^T \boldsymbol{H}_p(\hat{\boldsymbol{\theta}}_1, \hat{\boldsymbol{\theta}}_s)[(\boldsymbol{\theta}_1, \boldsymbol{\theta}_s) - (\hat{\boldsymbol{\theta}}_1, \hat{\boldsymbol{\theta}}_s)]$$

$$= (\boldsymbol{\theta}_1 - \hat{\boldsymbol{\theta}}_1)^T \boldsymbol{H}_{11}(\boldsymbol{\theta}_1 - \hat{\boldsymbol{\theta}}_1) + (\boldsymbol{\theta}_s - \hat{\boldsymbol{\theta}}_s)^T \boldsymbol{H}_{ss}(\boldsymbol{\theta}_s - \hat{\boldsymbol{\theta}}_s) + (\boldsymbol{\theta}_s - \hat{\boldsymbol{\theta}}_s)^T \boldsymbol{H}_{s1}(\boldsymbol{\theta}_1 - \hat{\boldsymbol{\theta}}_1) + (\boldsymbol{\theta}_1 - \hat{\boldsymbol{\theta}}_1)^T \boldsymbol{H}_{1s}(\boldsymbol{\theta}_s - \hat{\boldsymbol{\theta}}_s)$$

$$= (\boldsymbol{\theta}_1 - \hat{\boldsymbol{\theta}}_1)^T \boldsymbol{H}_{11}(\boldsymbol{\theta}_1 - \hat{\boldsymbol{\theta}}_1) + (\boldsymbol{\theta}_s - \hat{\boldsymbol{\theta}}_s)^T \boldsymbol{H}_{ss}(\boldsymbol{\theta}_s - \hat{\boldsymbol{\theta}}_s) + (\boldsymbol{\theta}_1 - \hat{\boldsymbol{\theta}}_1)^T (\boldsymbol{H}_{1s} + \boldsymbol{H}_{s1}^T)(\boldsymbol{\theta}_s - \hat{\boldsymbol{\theta}}_s).$$

Let us define $\boldsymbol{u} = \boldsymbol{\theta}_1 - \hat{\boldsymbol{\theta}}_1$, $\boldsymbol{v} = \boldsymbol{\theta}_s - \hat{\boldsymbol{\theta}}_s$ and $\boldsymbol{w} = \boldsymbol{H}_{11}^{-1}\boldsymbol{H}_{1s}\boldsymbol{v}$. We then have

$$
\begin{aligned}
(\boldsymbol{u}+\boldsymbol{w})^T \boldsymbol{H}_{11}(\boldsymbol{u}+\boldsymbol{w}) &= \boldsymbol{u}^T \boldsymbol{H}_{11}\boldsymbol{u} + \boldsymbol{u}^T \boldsymbol{H}_{11}\boldsymbol{w} + \boldsymbol{w}^T \boldsymbol{H}_{11}\boldsymbol{w} + \boldsymbol{w}^T \boldsymbol{H}_{11}\boldsymbol{u} \\
&= (\boldsymbol{\theta}_1 - \hat{\boldsymbol{\theta}}_1)^T \boldsymbol{H}_{11}(\boldsymbol{\theta}_1 - \hat{\boldsymbol{\theta}}_1) + (\boldsymbol{\theta}_1 - \hat{\boldsymbol{\theta}}_1)^T \boldsymbol{H}_{11}\boldsymbol{H}_{11}^{-1}\boldsymbol{H}_{1s}(\boldsymbol{\theta}_s - \hat{\boldsymbol{\theta}}_s) \\
&\quad + \boldsymbol{v}^T \boldsymbol{H}_{1s}^T \boldsymbol{H}_{11}^{-1}\boldsymbol{H}_{11}\boldsymbol{H}_{1s}\boldsymbol{v} + \boldsymbol{v}^T \boldsymbol{H}_{1s}^T \boldsymbol{H}_{11}^{-1}\boldsymbol{H}_{11}(\boldsymbol{\theta}_1 - \hat{\boldsymbol{\theta}}_1) \\
&= \boldsymbol{A} - \boldsymbol{v}^T \boldsymbol{H}_{ss}\boldsymbol{v} + \boldsymbol{v}^T \boldsymbol{H}_{1s}^T \boldsymbol{H}_{11}^{-1}\boldsymbol{H}_{1s}\boldsymbol{v} \\
&= \boldsymbol{A} - \boldsymbol{v}^T (\boldsymbol{H}_{ss} - \boldsymbol{H}_{1s}^T \boldsymbol{H}_{11}^{-1}\boldsymbol{H}_{1s})v \\
&= \boldsymbol{A} - \boldsymbol{v}^T \boldsymbol{\Omega}\boldsymbol{v},
\end{aligned}
$$

with $\boldsymbol{\Omega} = \boldsymbol{H}_{ss} - \boldsymbol{H}_{1s}^T \boldsymbol{H}_{11}^{-1}\boldsymbol{H}_{1s}$.

Thus

$$
\boldsymbol{A} = (\boldsymbol{u} + \boldsymbol{H}_{11}^{-1}\boldsymbol{H}_{1s}v)^T \boldsymbol{H}_{11}(\boldsymbol{u} + \boldsymbol{H}_{11}^{-1}\boldsymbol{H}_{1s}v) + \boldsymbol{v}^T \boldsymbol{\Omega}\boldsymbol{v}. \tag{10}
$$

Given equation (10), we are now able to prove Lemma 2, as

$$
\begin{aligned}
\int e^{l_p(\boldsymbol{\theta}_1, \boldsymbol{\theta}_s)} d\boldsymbol{\theta}_1 &= \int e^{l_p(\hat{\boldsymbol{\theta}}_1, \hat{\boldsymbol{\theta}}_s) - \frac{1}{2}\boldsymbol{A}} d\boldsymbol{\theta}_1 \\
&= \int e^{l_p(\hat{\boldsymbol{\theta}}_1, \hat{\boldsymbol{\theta}}_s)} e^{-\frac{1}{2}\boldsymbol{A}} d\boldsymbol{\theta}_1 \\
&= e^{l_p(\hat{\boldsymbol{\theta}}_1, \hat{\boldsymbol{\theta}}_s)} \int e^{-\frac{1}{2}\boldsymbol{A}} d\boldsymbol{\theta}_1 \\
&= e^{l_p(\hat{\boldsymbol{\theta}}_1, \hat{\boldsymbol{\theta}}_s)} \int e^{-\frac{1}{2}((\boldsymbol{u}+\boldsymbol{H}_{11}^{-1}\boldsymbol{H}_{1s}v)^T \boldsymbol{H}_{11}(\boldsymbol{u}+\boldsymbol{H}_{11}^{-1}\boldsymbol{H}_{1s}v) + \boldsymbol{v}^T \boldsymbol{\Omega}\boldsymbol{v})} d\boldsymbol{\theta}_1 \\
&= e^{l_p(\hat{\boldsymbol{\theta}}_1, \hat{\boldsymbol{\theta}}_s)} \int e^{-\frac{1}{2}((\boldsymbol{u}+\boldsymbol{H}_{11}^{-1}\boldsymbol{H}_{1s}v)^T \boldsymbol{H}_{11}(\boldsymbol{u}+\boldsymbol{H}_{11}^{-1}\boldsymbol{H}_{1s}v))} e^{-\frac{1}{2}\boldsymbol{v}^T \boldsymbol{\Omega}\boldsymbol{v}} d\boldsymbol{\theta}_1 \\
&= e^{l_p(\hat{\boldsymbol{\theta}}_1, \hat{\boldsymbol{\theta}}_s) - \frac{1}{2}\boldsymbol{v}^T \boldsymbol{\Omega}\boldsymbol{v}} \int e^{-\frac{1}{2}(\boldsymbol{\theta}_1 - \boldsymbol{z})^T \boldsymbol{H}_{11}(\boldsymbol{\theta}_1 - \boldsymbol{z})} d\boldsymbol{\theta}_1 \\
&= e^{l_p(\hat{\boldsymbol{\theta}}_1, \hat{\boldsymbol{\theta}}_s) - \frac{1}{2}\boldsymbol{v}^T \boldsymbol{\Omega}\boldsymbol{v}} (2\pi)^{\frac{p_1}{2}} |\det(\boldsymbol{H}_{11}^{-1})|^{\frac{1}{2}} (2\pi)^{-\frac{p_1}{2}} |\det(\boldsymbol{H}_{11}^{-1})|^{-\frac{1}{2}} \int e^{-\frac{1}{2}(\boldsymbol{\theta}_1 - \boldsymbol{z})^T \boldsymbol{H}_{11}(\boldsymbol{\theta}_1 - \boldsymbol{z})} d\boldsymbol{\theta}_1 \\
&= e^{l_p(\hat{\boldsymbol{\theta}}_1, \hat{\boldsymbol{\theta}}_s) - \frac{1}{2}\boldsymbol{v}^T \boldsymbol{\Omega}\boldsymbol{v}} (2\pi)^{\frac{p_1}{2}} |\det(\boldsymbol{H}_{11}^{-1})|^{\frac{1}{2}},
\end{aligned}
$$

where we re-arranged the terms so that the integral is over a normal distribution with mean $\boldsymbol{z} = \hat{\boldsymbol{\theta}}_1 - \boldsymbol{H}_{11}^{-1}\boldsymbol{H}_{1s}(\boldsymbol{\theta}_s - \hat{\boldsymbol{\theta}}_s)$ and covariance matrix $\boldsymbol{H}_{11}^{-1}$, which can be computed in closed form. $\square$

From Lemma 1 and Lemma 2, we can obtain equation (3) by replacing the denominator with the closed form above and taking the log on both size of equation (1). This yields

$$
\begin{aligned}
\log p(\boldsymbol{\theta}|\mathcal{D}) &\propto \log p(\mathcal{D} \mid \boldsymbol{\theta}_2, \boldsymbol{\theta}_s) + \log p(\boldsymbol{\theta}_1, \boldsymbol{\theta}_s) + \log p(\boldsymbol{\theta}_2, \boldsymbol{\theta}_s) - \log\left\{\int p(\mathcal{D} \mid \boldsymbol{\theta}_1, \boldsymbol{\theta}_s)p(\boldsymbol{\theta}_1, \boldsymbol{\theta}_s)d\boldsymbol{\theta}_1\right\} \\
&= \log p(\mathcal{D} \mid \boldsymbol{\theta}_2, \boldsymbol{\theta}_s) + \log p(\boldsymbol{\theta}_1, \boldsymbol{\theta}_s) + \log p(\boldsymbol{\theta}_2, \boldsymbol{\theta}_s) - l_p(\hat{\boldsymbol{\theta}}_1, \hat{\boldsymbol{\theta}}_s) + \frac{1}{2}\boldsymbol{v}^T \boldsymbol{\Omega}\boldsymbol{v} \\
&\propto \log p(\mathcal{D} \mid \boldsymbol{\theta}_2, \boldsymbol{\theta}_s) + \log p(\boldsymbol{\theta}_2, \boldsymbol{\theta}_s) + \log p(\boldsymbol{\theta}_1, \boldsymbol{\theta}_s \mid \mathcal{D}) + \frac{1}{2}\boldsymbol{v}^T \boldsymbol{\Omega}\boldsymbol{v}.
\end{aligned}
$$

## C  PLOTS FOR CNN SEARCH

In our CNN search experiment, we search for a"micro" cell as in (Pham et al., 2018). We employ the hyper-parameters available in the released ENAS code. The plots depicting error difference as a function of training epochs as provided in Figure 5 *(a)*, *(b)*and *(c)*. Note that here again the original ENAS is subject to multi-model forgetting, and our WPL helps reducing it. In Figure 5 *(d)*, we show the mean reward as training progresses. While the shape of the reward curve is different from the RNN case, because of a different formulation of the reward function, the general trend is the same; Our approach initially produces lower rewards, but is better at maintaining good models until the end of the search, as indicated by higher rewards in the second half of training.

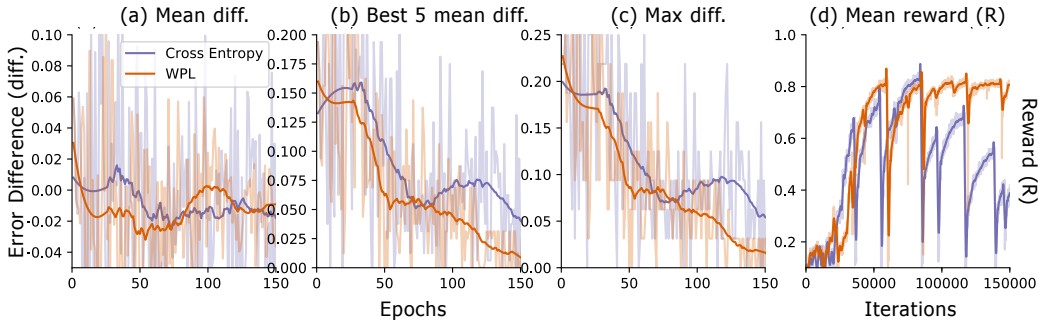

Figure 5: **Error differences when searching for CNN architectures.** Quantitatively, the multi-model forgetting effect is reduced by up to 99% for *(a)*, 96% for *(b)*, and 98% for *(c)*.

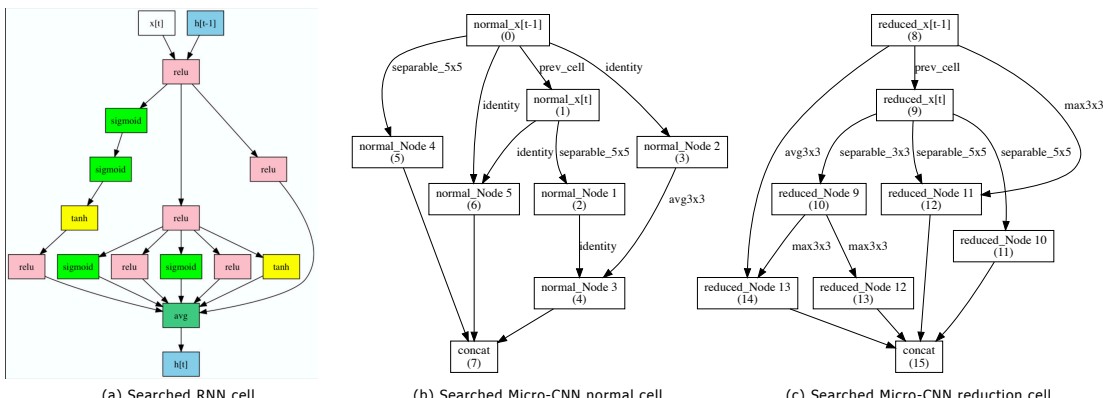

Figure 6: **Best architectures found for RNN and CNN.** We display the best architecture found by ENAS+WPL, in *(a)* for the RNN cell, and in *(b)* and *(c)* for the CNN normal and reduction cells.

## D    BEST ARCHITECTURES FOUND BY THE SEARCH

In Figure 6, we show the best architectures found by our neural architecture search for the RNN and CNN cases.

