# OpenReview forum: "Overcoming Multi-model Forgetting"
_ICLR.cc/2019/Conference_

### Official Review · AnonReviewer1 · 2018-11-01
**a good work on one shot model training**

**Rating:** 6
**Confidence:** 4

**Review:**

This paper discusses the phenomena of “neural brainwashing”, which refers to that the performance of one model is affected via another model sharing model parameters. To solve the issue, the authors derived a new loss out from maximizing the posterior of the parameters. With the new loss, the neural brainwashing is largely diminished.
The derived new loss looks meaningful to me and I think this is a valuable work for handling the weights coadaptation between two neural models, which with no doubt will bring great interests within the community of neural architecture search.

Here are some comments on the aspects that this paper can be improved:

1)	A very important related work [1] is missed in this paper. [1] discussed the properties of “one-shot model”, which means that several different architectures are unified into the same model by sharing model weights. Furthermore, [1] discussed “neural brainwashing” (although not with the same name) and how to handle it in a very simple way (by randomly dropping path). This definitely should be a baseline to compare with. In addition, a very recent work [2] also leverages model sharing to conduct neural architecture search.

2)	Although I understand that to improve accuracy of NAS is not the main goal of this paper, the baseline number to be improved over is too weak. For example, 4.87 of CIFAR10 in ENAS. Per my own hands on experience, it does not need too many hyperparameter tuning of ENAS to obtain < 4% error rate. Please provide more convincing baseline numbers and supporting evidences of the better performance of WPL in NAS.


[1] Bender, Gabriel, et al. "Understanding and simplifying one-shot architecture search." International Conference on Machine Learning. 2018.
[2] Luo, Renqian, et al. "Neural architecture optimization." NIPS (2018).

---

> ### Author Response · Authors · 2018-11-23
> **Response to Reviewer #1**
>
> Thank you for your time and positive feedback! We believe that addressing your comments has led to a stronger version of our paper.
>
> >Relation to [1] Bender et al., ICML 2018
>
> Thank you for pointing us to this interesting work. [1] indeed highlights the problems arising from training a one-shot model corresponding to multiple architectures with shared parameters, and circumvents them by randomly dropping paths during training.  However, this differs significantly from our work, where we derive a mathematical solution to address this problem. In fact, both solutions can be used jointly, and this is what is done in our experiments. Indeed, in all our architecture search experiments, ENAS relies on path dropout with a probability of 0.5, both when incorporating WPL and when not. Therefore, our experiments show that our approach can further improve the results of the strategy used in [1]. We have revised our paper so as to explain our use of path dropout and give proper credit to [1].
>
>
> > Relation to [2] Luo et al., NIPS 2018 (NAO)
>
> We became aware of NAO shortly after the ICLR deadline. In essence, the contribution of this work is to replace the reinforcement learning portion of ENAS with a gradient based auto-encoder. This can still suffer from multi-model forgetting, and again is thus orthogonal to our work. To demonstrate this, we incorporated WPL in the NAO framework and re-ran our RNN experiments with this new search method. The details of these experiments are provided in the appendix of our revised paper. These experiments illustrate the effectiveness of our approach with respect to both [1] and [2]. In short, we observed that
> the use of WPL reduces multi-model forgetting  in NAO, as in ENAS, and this for various dropout rates;
> while increasing the dropout rate indeed limits the multi-model forgetting effect, the resulting model consistently benefits from using WPL.
> We believe that these experiments confirm that our paper addresses an important issue, occurring in many neural architecture search strategies that use shared model representations. This further strengthens our contribution.
>
>
> > Relatively low performance of ENAS on CIFAR-10
>
> To implement our approach, we used a publicly available PyTorch implementation of ENAS for the RNN case that we further developed. For CIFAR-10, we extended this implementation to the CNN case. The choice of reimplementation of ENAS  was motivated by the simplicity and flexibility of PyTorch.
>
> To evaluate the final cells obtained by ENAS-WPL and ENAS we trained them in a fair training, without any hyperparameter tuning. The mismatch in scores is solely due to a difference in hyperparameter tuning, both for search and training from scratch, since ENAS final training is highly optimized. However, we believe this not to be a real issue, since our point is truly to demonstrate the benefits of accounting for multi-model forgetting, which our experiments do.

---

### Official Review · AnonReviewer3 · 2018-11-03
**Not sufficiently novel**

**Rating:** 5
**Confidence:** 5

**Review:**


- This "neural brainwashing" is catastrophic forgetting. Technically speaking, this is catastrophic forgetting.

- Also, some works targeting NAS (which I reckon should as well be cited due to being quite related) have targeted similar forgetting issues, e.g. Xu and Zhu, NIPS 2018 "Reinforced continual learning". It is nice to enrich the literature with new terms, when there is a need to. In my opinion, in this particular case, neural brainwashing is catastrophic forgetting.

- Forgetting is not necessarily an "individual problem", sticking to the language used in the third paragraph of the first page. The same applies to "single-model forgetting".

- page 1 "Our work is the first of which we are aware to identify neural brainwashing and to propose a solution.": According to the authors' argument, this is the case. Again, mine is different.

- Novelty w.r.t. works tackling catastrophic forgetting, most notably EWC, is minimal. Also, comparing to other state-of-the-art algorithms targeting catastrophic forgetting can further enrich the experiments.

- 3.1.1. On a technical level, there is no inherent difference, between EWC and the proposed algorithm.

- Writing can improve, both in terms of the flow and the language. There are also a few typos, e.g. in the first line of the caption of Figure 2.

- Apart from the aforementioned issue (comparing to other state-of-the-art catastrophic forgetting algorithms), the experiments are rigorously prepared.

---

> ### Author Response · Authors · 2018-11-23
> **Response to Reviewer #3**
>
> We thank the reviewer for their useful comments and for taking the time to review our paper. Below, we address their main concerns and have revised our paper accordingly.
>
> >Branding: Brainwashing vs forgetting
>
> In the literature, “forgetting” traditionally refers to the scenario where one aims to train a single model on two different datasets. By contrast, we aim to train multiple models on a single data, which motivated our use of the term “brainwashing”. We agree with the reviewer, however, that the term “forgetting” has recently started being used in a looser sense. Therefore, we have revised our paper to refer to our approach as “multi-model forgetting”.
>
>
> >Novelty over EWC
>
> There is clear technical novelty in our paper, which stems from the fact that, while EWC aims to maximize the posterior probability p(\theta | D1, D2), we maximize p(\theta_1, \theta_2, \theta_s | D), where \theta_1 and \theta_2 denote the parameters specific to each model and \theta_s those shared by both models. Heuristically modifying the EWC loss to fit our two-model scenario would be mathematically unjustified, and we therefore had to derive the equations for our formalism so as to reach WPL loss.
>
> Our derivation led to a new term in Equation (3), v^T \Omega v, which encodes the interaction between the two models. This term will never appear in EWC, nor in any single-model forgetting formulation. The fact that WPL looks similar to EWC loss is then only due to our use of a Laplace approximation of this term with the diagonal Fisher information matrix as covariance. However, other approximations, such as a Laplace one with a full covariance matrix, will lead to loss functions that differ fundamentally from the EWC one. We have clarified this in the revised paper and believe our mathematical formulation of the parameter sharing scenario and its general solution in Equation (3) to be solid technical contributions.
>
>
> >Relation to Xu & Zhu, NIPS 2018.
>
> This paper addresses a fundamentally different problem from the one we tackle. In essence, given Model A trained on Dataset A, Xu & Zhu, NIPS 2018, use an NAS-like strategy to train Model B on a different Dataset B. While Model B shares some parameters with Model A, absolutely no forgetting occurs, because the parameters of Model A are fixed. As such, this work does not address forgetting, but rather aims to compensate for the sub-optimality of Model A’s parameters for Dataset B via NAS. While interesting, this idea is orthogonal to ours. In fact, this method could benefit from relying on WPL when searching for the best Model B. We thank the reviewer for pointing us to this work, which we now discuss in our revised paper.

---

### Official Review · AnonReviewer2 · 2018-11-06
**The technique in this paper feels more or less identical to the ideas from Kirkpatrick et al (catastrophic forgetting). The difference seems to be one of application (different tasks vs same task), and as such feels like an incremental advance.**

**Rating:** 6
**Confidence:** 2

**Review:**

There is certainly additional novelty in that this paper focuses on models performing same/identical tasks (compared the results from the catastrophic forgetting paper), and because this model more clearly delineates the parameters that are shared across the models, vs those that are not. But both of those advances feel incremental.

---

> ### Author Response · Authors · 2018-11-23
> **Response to Reviewer #2**
>
> We thank the reviewer for taking the time to review our paper.  Below, we address the main concern.
>
> > Incremental advances
>
> Our work is not incremental. While weight sharing is popular, multi-model forgetting has been neither explicitly acknowledged, nor carefully studied. There is clear technical novelty in our paper, which stems from the fact that, while EWC aims to maximize the posterior probability p(\theta | D1, D2), we maximize p(\theta_1, \theta_2, \theta_s | D), where \theta_1 and \theta_2 denote the parameters specific to each model and \theta_s those shared by both models. Heuristically modifying the EWC loss to fit our two-model scenario would be mathematically unjustified, and we therefore had to derive the equations for our formalism so as to reach WPL loss.
>
> Our derivation led to a new term in Equation (3), v^T \Omega v, which encodes the interaction between the two models. This term will never appear in EWC, nor in any single-model forgetting formulation. The fact that WPL looks similar to EWC loss is then only due to our use of a Laplace approximation of this term with the diagonal Fisher information matrix as covariance. However, other approximations, such as a Laplace one with a full covariance matrix, will lead to loss functions that differ fundamentally from the EWC one. We have clarified this in the revised paper and believe our mathematical formulation of the parameter sharing scenario and its general solution in Equation (3) to be solid technical contributions.
>
> Furthermore, we believe that our new experiments using WPL in the NAO framework of Luo et al., NIPS 2018,  further confirm that our paper addresses an important issue that occurs in many neural architecture search strategies that use shared model representations.

---

### Public Comment · ~Vihari_Piratla1 · 2018-09-29
**Brainwashing vs Catastrophic Forgetting**

The difference between what is referred to as "Neural Brainwashing" and "Catastrophic Forgetting" is not clear from the explanation. Even in *Catastrophic Forgetting*, the performance of the network on earlier tasks degrades. Also not clear is how the proposed model is different from the regularization methods (akin to "weight plasticity" proposed in yours) suggested in the context of Catastrophic Forgetting.

---

> ### Author Response · Authors · 2018-09-29
> **brainwashing is different from catastrophic forgetting**
>
> We thank the reader for commenting on the paper so quickly. We believe that the problem statement for brainwashing is very different from the forgetting. Throughout the paper (e.g.,  paragraph 2 of the Introduction, Section 3.1.1), we explain this difference, i.e., catastrophic forgetting happens when training a single model for multiple tasks, while the brainwashing occurs when training  multiple models on a single task.
>
> Indeed, In « overcoming catastrophic forgetting », the authors maximize the posterior probability p(\theta | D1, D2) while in « overcoming neural brainwashing », we maximize p(\theta_1, \theta_2, \theta_s | D), with \theta_s refering to the shared parameters between two different models. Mathematically speaking, the two problems are fundamentally different. Catastrophic forgetting does not deal with parameter sharing across different models and only considers a single model with parameters \theta.  In our paper, we focus on tackling the problem where multiple models are sharing part of their architectures. We formulate our final loss through a completely different mathematical derivation and coincidentally ends in a similar formalism.

---

### Author Response · Authors · 2018-11-23
**Revised paper uploaded**

We thank the reviewers for their valuable comments and for taking the time to review our paper. We have uploaded a revised version that addresses the reviewers’ main concerns. In particular

- We have renamed brainwashing “multi-model forgetting” to account for the fact that the literature has become more liberal in using the term “forgetting”;

- We have clarified the novelty of our approach. Specifically, one could not simply heuristically modify the EWC loss to fit our multi-model setting. Our derivation shows that an additional term encoding the interactions between two models arises from our scenario;

- We have added a discussion of (Xu & Zhu, NIPS 2018);

- We have incorporated novel experiments based on the NAO strategy of Luo et al., NIPS 2018. These experiments demonstrate the benefits of our WPL loss in another neural architecture search approach, and show that WPL improves over the path dropout strategy of Bender et al., ICML 2018.


Altogether, we believe that these modifications significantly strengthen our paper and further highlight the generality of our approach.

---

### Meta-Review · Area_Chair1 · 2018-12-13
**good work but more is needed to have impact**

**Confidence:** 3
**Recommendation:** Reject

**Metareview:**


pros:
- nicely written paper
- clear and precise with a derivation of the loss function

cons:

novelty/impact:
I think all the reviewers acknowledge that you are doing something different in the neural brainwashing (NB) problem than is done in the typical catastropic forgetting (CF) setting.  You have one dataset and a set of models with shared weights; the CF setting has one model and trains on different datasets/tasks.  But whereas solving the CF problem would solve a major problem of continual machine learning, the value of solving the NB problem is harder to assess from this paper...  The main application seems to be improving neural architecture search.  At the meta-level, the techniques used to derive the main loss are already well known and the result similar to EWC, so they don't add a lot from the analysis perspective.  I think it would be very helpful to revise the paper to show a range of applications that could benefit from solving the NB problem and that the technique you propose applies more broadly.